# DspA/E-Triggered Non-Host Resistance against *E. amylovora* Depends on the Arabidopsis *GLYCOLATE OXIDASE 2* Gene

**DOI:** 10.3390/ijms23084224

**Published:** 2022-04-11

**Authors:** Alban Launay, Sylvie Jolivet, Gilles Clément, Marco Zarattini, Younes Dellero, Rozenn Le Hir, Mathieu Jossier, Michael Hodges, Dominique Expert, Mathilde Fagard

**Affiliations:** 1Université Paris-Saclay, INRAE, AgroParisTech, Institut Jean-Pierre Bourgin (IJPB), 78000 Versailles, France; alban.launay@gmail.com (A.L.); sylvie.jolivet@inrae.fr (S.J.); gilles.clement@inrae.fr (G.C.); marco.zarattini@ulb.be (M.Z.); rozenn.le-hir@inrae.fr (R.L.H.); dominique.expert@inrae.fr (D.E.); 2Université Paris-Saclay, CNRS, INRAE, Université Evry, Institute of Plant Sciences Paris-Saclay (IPS2), 91190 Gif sur Yvette, France; younes.dellero@inrae.fr (Y.D.); mathieu.jossier@universite-paris-saclay.fr (M.J.); michael.hodges@universite-paris-saclay.fr (M.H.); 3Université Paris Cité, CNRS, INRAE, Institute of Plant Sciences Paris-Saclay (IPS2), 91190 Gif sur Yvette, France

**Keywords:** *Arabidopsis thaliana*, *Erwinia amylovora*, type II non-host resistance, type three effector, DspA/E, glycolate oxidase, peroxisome, photorespiration

## Abstract

DspA/E is a type three effector injected by the pathogenic bacterium *Erwinia amylovora* inside plant cells. In non-host *Arabidopsis thaliana*, DspA/E inhibits seed germination, root growth, de novo protein synthesis and triggers localized cell death. To better understand the mechanisms involved, we performed EMS mutagenesis on a transgenic line, 13-1-2, containing an inducible *dspA/E* gene. We identified three suppressor mutants, two of which belonged to the same complementation group. Both were resistant to the toxic effects of DspA/E. Metabolome analysis showed that the 13-1-2 line was depleted in metabolites of the TCA cycle and accumulated metabolites associated with cell death and defense. TCA cycle and cell-death associated metabolite levels were respectively increased and reduced in both suppressor mutants compared to the 13-1-2 line. Whole genome sequencing indicated that both suppressor mutants displayed missense mutations in conserved residues of Glycolate oxidase 2 (GOX2), a photorespiratory enzyme that we confirmed to be localized in the peroxisome. Leaf GOX activity increased in leaves infected with *E. amylovora* in a DspA/E-dependent manner. Moreover, the *gox2-2* KO mutant was more sensitive to *E. amylovora* infection and displayed reduced JA-signaling. Our results point to a role for glycolate oxidase in type II non-host resistance and to the importance of central metabolic functions in controlling growth/defense balance.

## 1. Introduction

During evolution, plants have developed a sophisticated and multi-layered immune system that enables them to perceive and counteract potential invaders. The basal line of defense relies on the recognition of structural microbial components, i.e., bacterial flagellin, fungal chitin and lipopolysaccharides, commonly designated as microbe- or pathogen- associated molecular patterns (MAMPs or PAMPs) [1], by specific cell surface-localized receptors named Pattern Recognition Receptors (PRRs) [2,3]. A classic example of a PAMP-PRR interaction is the recognition of the conserved twenty-two amino acid epitope of flagellin, defined as FLG22, by the leucine-rich repeat receptor-like kinases (LRR-RLKs) Flagellin-sensitive 2 (FLS2) [4]. This line of defense, defined as PAMP-Triggered Immunity (PTI), consists in a complex signaling network mediated by mitogen-activated protein kinases (MAPK), calcium-dependent protein kinases (CPK or CDPK) as well as phytohormones such as salicylic acid (SA), jasmonic acid (JA) and ethylene (ET). PTI activation leads to a set of induced responses that reinforce pre-formed defenses [5,6]. Callose deposition, cell wall modifications along with a rapid increase in cytosolic [Ca^2+^] and reactive oxygen species (ROS) accumulation are common responses triggered during PTI [7,8,9]. Although PTI ensures protection against a wide range of pathogens, successful pathogens can overcome this first line of defense by injecting inside the plant cell effector molecules interfering with PTI activation [10,11]. In response, resistant plants deploy intracellularly localized receptors called nucleotide-binding leucine-rich repeat receptors (NLRs), encoded by R genes. The recognition of virulent effectors leads to the activation of the second layer of immunity defined as Effector Triggered Immunity (ETI) [12]. ETI is generally stronger than PTI in magnitude and is associated with increased expression of defense genes and hypersensitive response (HR), culminating into a programmed cell death (PCD) at the site of infection [10,13]. Moreover, PTI and ETI are closely interconnected, require common signaling components and mutually potentiate each other [14,15]. ETI has been found to be effective almost exclusively against biotrophic and hemibiotrophic pathogens but not against necrotrophic pathogens [16].

When a plant species is entirely resistant to all genetic variants of a pathogen, this is defined as non-host resistance. Although the underlying molecular mechanisms are not fully understood, it is considered as one of the most robust forms of resistance. This defense strategy is regulated by multiple genes, and it is multi-layered, often involving both PTI and ETI activation [17,18,19,20]. Based on the absence or presence of HR, non-host resistance is classified into type I or type II, respectively [18]. The more complex HR-dependent type II non-host resistance is triggered when pathogens can overcome basal plant defenses. In HR activation, a crucial role is played by the bacterial type III secretion system (T3SS) which translocates an array of proteinaceous type three effectors (T3Es), directly into the plant cell cytosol [21]. So far, many T3Es from several gram-negative bacteria were reported to modulate PTI- and ETI-associated immune responses, phytohormone signaling, plant gene expression and cell death [22]. On the other hand, type II non-host resistance resembles the incompatible gene-for-gene interaction theory, by which, via R genes, resistant plants can recognize T3Es that trigger HR and cell death [20].

*Erwinia amylovora* is a bacterial necrotrophic plant pathogen causing the fire blight disease of apple, pear and other rosaceous plants. The pathogenicity of *E. amylovora* strongly depends on the intracellular delivery of T3Es by T3SS. Although at least four T3Es are deployed by *E. amylovora* during plant infection, only T3E DspA/E is fundamental for pathogenicity, as a mutant strain lacking the DspA/E effector is non-pathogenic [23]. Furthermore, DspA/E-dependent *E. amylovora* growth in host plants is associated with the triggering of necrosis and ROS accumulation, suggesting a role for DspA/E-triggered cell death in disease. When expressed directly in plant cells, both host and non-host, either transiently [24] or in transgenic plants [25], DspA/E triggers cell death evoking the HR and activates defense. We previously generated and characterized *A. thaliana* transgenic plants expressing DspA/E under an estradiol-inducible promoter [25]. Understanding the mechanisms underlying DspA/E cell death and its role during the infection process could lead to the identification of new targets for plant protection against *E. amylovora*. Here, we performed ethyl methanesulfonate (EMS)-mediated mutagenesis of the DspA/E-expressing *A. thaliana* 13-1-2 line and we identified and characterized DspA/E suppressor mutants. We show that two allelic mutants display non-synonymous mutations in the *Glycolate Oxidase 2* (*GOX2*) gene, encoding a glycolate oxidase enzyme involved in photorespiration and non-host resistance [26,27]. We show that these point mutations in *GOX2* suppress DspA/E-associated phenotypes. Interestingly, the mutations harbored by the two suppressor lines led to a decrease in GOX activity using both in vitro purified recombinant proteins and in vivo leaf soluble protein extracts. We discuss the role of GOX2 in DspA/E-triggered cell death and in non-host resistance and the links between these processes.

## 2. Results

### 2.1. Identification of DspA/E Suppressor Mutants

We previously generated and characterized two transgenic lines, 13-1-1 and 13-1-2, expressing DspA/E under the control of an estradiol-inducible promoter [25]. Estradiol-induction of DspA/E in *A. thaliana* is toxic for plants (Figure 1a) and ultimately leads to plant cell death. In order to identify the mechanism(s) by which DspA/E triggers plant cell death, we subjected line 13-1-2 to EMS, which induces random point mutations in the genome. Approximately 125,000 M0 seeds of 13-1-2 were treated overnight with either 0.2% or 0.4% EMS, rinsed in water, sterilized and grown in vitro for two weeks. Viable seedlings were transferred to soil and the M2 progeny was recovered (Figure 1b). M2 seedlings were sown in vitro on 10 nM estradiol, a condition in which original 13-1-2 seeds do not germinate due to DspA/E toxicity (Figure 1a and Figure 2a). Each putative mutant was backcrossed to the parental line and the progeny was sown on estradiol to identify recessive mutations (Appendix A).

### 2.2. Phenotypical Characterization of Suppressor Mutants

We identified three putative recessive mutants (I-18, II-36 and I-40) in which the DspA/E transgene was expressed at a level similar to the parental 13-1-2 line (Figure 2b). The three mutants were crossed with each other, and progeny were tested for their resistance to DspA/E. The progeny of crosses from mutants I-18 and II-36 were all resistant to DspA/E toxicity since they germinated on estradiol (Appendix A), indicating that these two mutations constitute a complementation group. The progeny of all crosses with the third mutant, I-40, were sensitive to estradiol, indicating that this mutant did not belong to the same complementation group as mutants I-18 and II-36 (Appendix A). The two mutants belonging to the same complementation group, I-18 and II-36, were selected for further analysis. 

The 13-1-2 transgenic line displays several phenotypes associated with the expression and the toxicity of DspA/E including inhibition of de novo protein synthesis and inhibition of root elongation [25]. In order to determine which aspects of DspA/E toxicity were affected in the two selected suppressor mutants, I-18 and II-36, we compared their phenotypes with that of the 13-1-2 parental line following estradiol-induced DspA/E expression. Root tips of one-week-old seedlings were treated with estradiol for 10 min and root growth was measured 24 h after treatment. As described previously, the 13-1-2 line showed a complete inhibition of root growth per day following the induction of DspA/E expression (Figure 3a). Following estradiol treatment, both suppressor-mutants showed root growth similar to mock-treated control conditions (without estradiol), indicating that these mutants were resistant to the toxicity of DspA/E for root growth (Figure 3a). We then analyzed the capacity of DspA/E to suppress de novo protein synthesis using 35S-labeled methionine. As previously described [25], when DspA/E expression was induced in the parental 13-1-2 line, de novo protein synthesis was stopped 3 h following the estradiol treatment (Figure 3b). In both suppressor mutants, de novo protein synthesis was not blocked following the induction of DspA/E expression, indicating that these mutants are resistant to the toxic effect of DspA/E on de novo protein synthesis. Altogether, our data show that DspA/E toxicity and associated cellular phenotypes are abolished in both suppressor mutants identified.

### 2.3. Identification of Causal Mutations in the Glyoclate Oxidase 2 Gene

In order to identify the causal mutations leading to suppression of DspA/E toxicity, we adapted the Shoremap strategy described previously [26]. For this, we crossed each suppressor mutant with the parental 13-1-2 line. The F1 progeny of this cross was then self-pollinated and the F2 progeny was analyzed to identify individuals resistant to DspA/E toxicity. After sowing on 100 nM estradiol, 100 F2 individuals that germinated on estradiol, and thus resistant to DspA/E toxicity, were sampled and pooled for genomic sequencing. This was performed in parallel for both the I-18 and the II-36 suppressor mutants.

Illumina sequencing of the pools of F2 individuals bearing the I-18 and II-36 suppressor mutations was performed at a 40× depth. Sequence analysis was performed using the CLC genomics workbench software. In parallel, the parental 13-1-2 line was sequenced in order to eliminate single nucleotide polymorphisms (SNPs) already present. We then identified the SNPs present in the F2 pool of each suppressor mutant. We found a region of chromosome 3 in which the percentage of SNPs in both suppressor mutants was between 80 and 100% (Appendix A). This region contained five candidate genes and detailed analysis of the SNPs present in both suppressor mutants showed that only one gene had SNPs leading to non-synonymous mutations in both suppressors; the At3g14415 gene encoding GLYCOLATE OXIDASE 2 (GOX2). In the II-36 mutant the identified SNP led to a Ser27Phe mutation in the GOX2 protein, while in the I-18 mutant the identified SNP led to an Ala96Val mutation (Figure 4a,b). A multiple sequence alignment analysis showed that both mutations altered conserved amino acids of GOX enzymes from different plant species, including a GOX enzyme from *Malus domestica* (Figure 4c).

### 2.4. The Knock-Out gox2-2 Mutant Is Allelic to the DspA/E Toxicity Suppressor Mutants

In order to confirm that mutations in the *GOX2* gene were responsible for the loss of DspA/E toxicity in *A. thaliana* seedlings, we crossed both the parental 13-1-2 line and the suppressor mutants with the *gox2-2* T-DNA KO line described previously [27]. When crossed with the parental 13-1-2 line, the F1 progeny were unable to germinate on estradiol, as expected since the *gox2-2* mutation is recessive. When the *gox2-2* mutant was crossed with the I-18 suppressor mutant, the F1 progeny was able to germinate on estradiol (Figure 5a). Similar results were obtained when the *gox2-2* mutant was crossed with the II-36 suppressor. Thus, the *gox2-2* KO mutant was unable to functionally complement the suppressor mutants for the susceptibility to DspA/E toxicity. This result is consistent with the fact that DspA/E toxicity in *A. thaliana* requires GOX2.

In *A. thaliana*, GOX enzymes, involved in photorespiration, have been attributed to a family of five genes; however, only three of them have been shown to convert glycolate to glyoxylate; GOX1-3 [28]. The GOX2 protein has been shown to be involved in non-host resistance [27], and this role is believed to be linked to ROS production since GOX enzymatic activity generates H_2_O_2_ [29,30]. We tested the global GOX activity in *A. thaliana* leaf extracts inoculated or not with *E. amylovora*. We found that in response to *E. amylovora* infection, there was an increase in the extractable GOX activity of leaf extracts (Figure 5b). Since *A. thaliana* rosettes contain both GOX1 and GOX2 [28], it was not possible to determine whether the observed increase in GOX activity is essentially due to GOX2 alone. However, smaller increase in GOX activity in the *gox2-2* following inoculation with *E. amylovora* could indicate that this is the case (Figure 5b). We then compared leaf GOX activity in wild-type plants and the *gox2-2*, I-18 and II-36 mutants. As expected, the *gox2-2* KO mutant showed a 47% reduction in GOX activity compared to the control wild-type line (Figure 5c). This was also true for the II-36 suppressor mutant in which a 30% reduction in GOX activity was observed. In the I-18 suppressor mutant the leaf GOX activity was 15% lower than in wild-type; however, this difference was not significant, suggesting that the I-18 mutation might not affect global GOX activity.

### 2.5. GOX2 Is Required to Induce JA-Dependent Defense and Confers Non-Host Resistance to E. amylovora

GOX2 has been previously shown to be involved in non-host resistance [27]. We therefore analyzed the response of the *gox2-2* mutant to *E. amylovora* infection. As expected, *gox2-2* was more susceptible to *E. amylovora* (Figure 6a). This was correlated with a reduction in electrolyte leakage in the mutant in response to *E. amylovora* infection (Figure 6b), which is consistent with a reduction in defense response reminiscent of type II non-host. We then analyzed the expression of two defense-signaling marker genes, *CHI-B* and *PR1*, both previously shown to be induced in wild-type plants in response to *E. amylovora* infection [19]. We found that the SA-dependent *PR1* gene was unaffected in the *gox2-2* mutant while the JA-dependent *CHI-B* gene was less induced in the *gox2-2* mutant background (Figure 6c). Furthermore, GOX enzymes catalyze reactions that oxidize different 2-hydroxy acid substrates and all produce H_2_O_2_. In the case of GOX2, H_2_O_2_ production occurs in response to infection by non-adapted pathogens [27]. However, we did not find any difference in response to *E. amylovora* in terms of H_2_O_2_ accumulation detected by DAB staining at the level of *gox2-2* leaves when compared to wild-type leaves (Figure 6d).

T3Es are injected into the plant cytoplasm; however, their final localization within the plant cell can be in the cytosol and/or in different organelles. Motif search in the DspA/E protein sequence showed the presence of several organelle-targeting signals including a peroxisome targeting signals (PTS) [31]. However, previous works trying to establish the intracellular localization of DspA/E were unsuccessful, probably due to the toxicity of this protein for eukaryotic cells [25,32,33]. Since GOX2 suppresses DspA/E toxicity, it was decided to check the subcellular localization of the GOX2 protein. Its revpresumed peroxisomal localization is based on the presence of a typical C-terminal tripeptide required to address proteins into peroxisomes [34] and its presence in peroxisomal proteomes [35] but its subcellular localization has not been confirmed by other means. We constructed a GOX2::GFP fusion under the native GOX2 promoter and transformed protoplasts obtained from a stable transgenic line bearing a peroxisomal RFP marker [36]. Our results confirm that GOX2 is indeed peroxisomal (Figure 7).

### 2.6. DspA/E-Triggered Cell Death Is Associated with Strong Metabolite Changes

In order to better understand the mechanisms underlying both DspA/E triggered cell death and its suppression by the two suppressor mutants, we performed a metabolomic analysis by GC-MS. Seedlings of the 13-1-2 parental line and of the two suppressor mutants were either mock-treated (without estradiol) or treated with increasing estradiol concentrations. The results show that DspA/E expression in the 13-1-2 parental line leads to depletion in metabolites associated with the TCA cycle, such as fumarate and citrate (Figure 8). In the 13-1-2 parental line an accumulation of metabolites associated with cell death and defense such as pipecolate and stigmasterol was observed (Figure 8). Other metabolites showed an altered accumulation in response to DspA/E expression such as salicylate and several organic acids (Figure 9). Interestingly, TCA cycle intermediates accumulated more in the suppressor mutants than in the 13-1-2 parental line. On the contrary, cell-death and defense-associated metabolites showed levels in the suppressor mutants comparable to the control line (Figure 8 and Figure 9).

Altogether, our data show that expression of DspA/E in *A. thaliana* seedlings leads to important modifications of the metabolome and that part of these modifications of metabolite levels in response to DspA/E expression were lost in both suppressor mutants (Figure 8 and Figure 9).

## 3. Discussion

The pathogenicity of *E. amylovora* depends on its T3SS which injects type three effectors (T3Es) inside the plant cell; among them, the T3E DspA/E plays a major role since a DspA/E-deficient mutant is non-pathogenic [23]. In host plants, DspA/E causes a rapid oxidative burst associated with disease [38,39]. In the model plant *A. thaliana*, *E. amylovora* is able to multiply transiently in a T3SS-dependent manner [40]. However, *A. thaliana* can restrict bacterial growth by triggering an active type II non-host resistance that includes callose deposition, cell death and expression of defense signaling pathway genes [19]. In a previous study, we generated the transgenic *A. thaliana* line 13-1-2 expressing *dspA/E* under an estradiol-inducible promoter and showed that DspA/E plays an important role in triggering non-host resistance in *A. thaliana* [25]. However, the exact mechanisms by which DspA/E triggered cell death in the line 13-1-2 remained to be elucidated.

In the present study, we EMS mutagenized the 13-1-2 line and identified two independent suppressor mutants, I-18 and II-36, that were resistant to DspA/E-inhibition of seedling germination. Further characterization of these two mutants showed that they were rescued for all phenotypes associated with DspA/E expression that were tested: inhibition of germination, inhibition of root elongation, and inhibition of de novo protein synthesis. To further understand the mechanisms at play, we performed a metabolomic analysis of the 13-1-2 line and of the two suppressor mutants (Figure 8). Interestingly, we observed that the TCA cycle was strongly affected in the 13-1-2 line when DspA/E was expressed, which could explain the lack of germination of transgenic seeds in these conditions [41,42]. In contrast, both suppressor mutants showed higher accumulation of TCA cycle intermediates such as fumarate, 2-oxoglutarate, malate and succinate (Figure 8), thus perhaps explaining their capacity to germinate while expressing DspA/E. Indeed, there is increasing evidence of that metabolites such as TCA cycle intermediates can act as signaling intermediates [43]. Another key feature of the metabolomic response of *A. thaliana* seedlings to the expression of DspA/E was the accumulation of several metabolites associated with cell death and defense. For example, pipecolate, a catabolite of lysine, that can induce resistance to *P. syringae* pv. *maculicola* [44], accumulated strongly in response to DspA/E expression. Stigmasterol, known to reduce membrane fluidity and permeability also accumulated strongly in response to DspA/E expression [45]; DspA/E expression triggers strong electrolyte leakage and it is possible that stigmasterol accumulation is a means for the plant to control this leakage. All of these metabolites accumulated less in the two suppressor mutants, indicating that these metabolites could play a key role in the signaling of DspA/E-triggered defense and/or in the coping with damage to the plant cells associated with DspA/E expression. 

Whole genome sequencing of the two suppressor mutants allowed us to determine that they each carried different missense mutations in the *A. thaliana GOX2* gene, which encodes a glycolate oxidase involved in photorespiration [28]. GOX2 belongs to a family of five genes: *GOX1*, *GOX2*, *GOX3*, *HAOX1* and *HAOX2* [30]. Previous analysis of single knock-out lines for GOX1 and GOX2 and an artificial microRNAi line targeting both genes in Arabidopsis showed that these two genes were the major photorespiratory genes in leaves [28]. Conversely, other works showed that GOX3 encoded for a lactate oxidase involved in roots metabolism during hypoxia [46], while HAOX1 and HAOX2 encoded for long-chain 2-hydroxy acid oxidases mainly expressed in seeds [47]. Nevertheless, all *GOX* genes seem to play a role in the non-host resistance against *P. syringae* pv. *tabaci* in Arabidopsis leaves, since *GOX3*, *HAOX1* and *HAOX2* were significantly induced after 24 h of infection with this pathogen [27]. Here, we showed that, following *E. amylovora* infection, overall leaf GOX activity can be significantly induced in a DspA/E-dependent manner in Col-0 plants while it was not the case in the *gox2-2* mutant (Figure 5b). Interestingly, the two suppressor lines harboring different mutations in the coding region of GOX2 genes had also a lower overall leaf GOX activity (Figure 5c).

GOX2 has been identified in the proteome of peroxisomes [34,35], and here confocal microscopy using a peroxisomal RFP marker and a GOX2::GFP construct allowed us to confirm that GOX2 is indeed a peroxisomal protein. Both suppressor mutations modified a different conserved amino acid of the GOX2 protein. Interestingly, GOX has already been shown to be involved in non-host resistance against *P. syringae* pv. *tabaci* [27]. In this study, ROS accumulation in the leaf blade in response to a non-adapted pathogen was lower in knock-out *gox* mutants. *GOX2* is part of a five member gene family comprising *GOX1*, *GOX2*, *GOX3*, *HAOX1*, and *HAOX2* and perhaps surprisingly all GOX family knock-out mutants exhibited a reduction in ROS accumulation in response to the non-adapted pathogen *P. syringae* pv. *tabaci* [27]. We found no major difference in macroscopic accumulation of ROS in the *gox2-2* background using the non-adapted pathogen *E. amylovora*, suggesting that in this case, there were GOX2-independent sources of ROS. Indeed, we showed previously that ROS production in response to *E. amylovora* in *A. thaliana* is strongly dependent on the RBOHD NADPH oxidase [48]. The KO *gox2-2* mutant showed a higher susceptibility to *E. amylovora*, showing the involvement of this enzyme in type II non-host resistance against a necrotrophic pathogen. Interestingly, we found normal expression of the SA-dependent *PR1* gene in the *gox2* mutant background but low expression of the JA-signaling marker gene *CHI-B*. Since JA biosynthesis is in part localized in the peroxisome [49], it could be speculated that perturbation of peroxisomal functions in the *gox2* mutant alters the capacity of the plant to synthesize JA and/or compromises JA signaling upon pathogen attack. Therefore, we show that in addition to being involved in non-host resistance against hemi-biotrophic pathogens, GOX2 is also involved in type II non-host resistance against necrotrophic pathogens.

In this study, we show that *A. thaliana gox2-2* mutants are resistant to the toxicity of DspA/E and more susceptible to *E. amylovora*. This suggests that, in the non-host context, DspA/E toxicity is correlated with defense activation. This is interesting as the role of the toxicity of DspA/E in plant cells during the infection process is still unclear to date. However, the precise role of GOX2 in both the toxicity of DspA/E and the susceptibility to *E. amylovora* remains to be determined. The *gox2-2* mutant showed reduced expression of *CHI-B*, a JA-signaling marker gene, indicating reduced defense activation in this genetic background, which could explain the increased susceptibility to *E. amylovora*. We also observed that in response that in response to *E. amylovora* infection there is an increase in total GOX activity, which was strongly reduced in the *gox2-2* mutant. This result suggests that the peroxisome could be targeted during the infection process. 

Finally, GOX2 is an enzyme involved in photorespiration, an energy-costly process, which can limit crop yield, especially under low CO_2_ conditions that can be produced by biotic-stress related stomatal closure. It is well known that there is a trade-off between growth and defense in plants [50]. In the presence of DspA/E and in response to *E. amylovora* infection, the growth-defense equilibrium could be altered. One explanation for the suppression of DspA/E toxicity by the mutation of the GOX2 enzyme could be an alteration of growth-defense equilibrium. In this scenario, loss of GOX2 and associated defense reactions could re-equilibrate the defense/growth balance in favor of growth. Forward genetics is a powerful strategy to uncover unexpected links between different cellular processes. Central metabolic functions and specialized metabolites are increasingly shown to play a crucial role in plant-pathogen interactions. Further studies will be necessary to uncover the precise links between these processes.

## 4. Materials and Methods

### 4.1. Plant Material and Growth Conditions

In vitro growth was performed on 1× Murashige and Skoog (MS) medium supplemented with 1% sucrose in growth chambers under a long day light regime (16 h light/8 h dark) at 25 °C (day) and 20 °C (night).

The 13-1-2 *A. thaliana* transgenic line, described previously, bears the *dspA/E* gene under the control of an estradiol-inducible promoter [25]. The *gox2-2* mutant bears a T-DNA insertion in the 5′UTR region of the AT3G14415 gene and has been shown to be a knock-out mutant [27]. All the plant material used in this study is in the Col-0 accession.

### 4.2. Bacterial Strains and Pathogen Infection

The bacterial strains used in this study are the wild-type CFBP1430 strain of *Erwinia amylovora* and the M81 mutant defective for DspA/E [25].

For pathogen infections, rosette leaves of 5-week-old plants were infiltrated with *E. amylovora* CFBP1430 or the M81 *dspA/E*- mutant using a needleless syringe. Bacterial suspensions were prepared in sterile water (10^7^ CFU/mL; OD_600_ = 0.1). Twenty-four hours post infection (hpi), we performed bacterial counting by grinding infected leaves using glass beads in a TissueLyser (Qiagen/Retsch, Hilden, Germany). The bacterial suspensions were used to prepare serial dilutions, which were plated on an LB medium, and after 1 or 2 days the colonies formed were counted to evaluate the initial number of bacteria.

### 4.3. EMS Mutagenesis and Screening

EMS mutagenesis of the 13-1-2 line was performed as described previously [51,52]. Briefly, approximately 125,000 seeds (2.5 g) were treated with 0.2 or 0.4% EMS overnight, rinsed in water, sterilized and grown in vitro for two weeks. Viable seedlings were transferred to soil in growth chambers (16 h light/8 h dark) at 25 °C (day) and 20 °C (night) and the M1 progeny was recovered in 150 bulks of approximately 10 plants each. The screening was performed by sowing on MS supplemented with 1% sucrose and 10 nM estradiol. Seedlings able to germinate in these conditions were transferred to fresh MS without estradiol for a few days and then transferred to soil to collect their progeny.

### 4.4. Genome Sequencing of Suppressor Mutants and Analysis

The selected suppressor mutants were backcrossed with the parental 13-1-2 line and the F2 progeny was analyzed. For each suppressor mutant, 100 F2 individuals able to germinate on 10 nM estradiol were sampled and pooled. For each pool, DNA was extracted and whole genome Illumina^®^ sequencing at a 40× depth was performed. The sequences were mapped to the *A. thaliana* Col-0 reference genome using the CLC genomics workbench software. In parallel, the parental line 13-1-2 was sequenced to identify the SNPs it carried, these SNPs were then eliminated from our analysis of the genomes of the suppressor mutants.

### 4.5. Recombinant GOX2 Production and Site-Directed Mutagenesis of AtGOX2

To produce recombinant GOX2 proteins, the previously described pET28a-AtGOX2 expression plasmid [53] was used as a template to introduce the desired point mutations by PCR using specific primer pairs (Appendix A) and the QuikChange^®^ II XL site-directed mutagenesis kit (Agilent^®^, Les Ulis, France), according to the manufacturer’s instructions. This strategy generated AtGOX2-S27F and AtGOX-A96V mutated proteins. All constructions were subsequently verified by DNA sequencing using T7 and T7-term primers (Appendix A).

The N-terminal His-tagged GOX2 proteins were purified by affinity chromatography as previously described [28]. The purity of each recombinant GOX2 protein was checked by SDS-PAGE (10% acrylamide) stained with Coomassie Brilliant Blue [54].

Recombinant GOX2 activities were measured using 5 µg of purified recombinant GOX2 in 50 mM Tris-HCl, 0.1 mM FMN, pH 8.0 and increasing glycolate (0.05 to 10 mM) concentrations by an enzyme-coupled reaction at 30°C. H_2_O_2_ produced by GOX activity was quantified in the presence of 0.4 mM O-dianisidine and 2 U horseradish peroxidase by measuring the ΔA440 nm with a Varian Cary 50 spectrophotometer. KM and activity values were calculated using SigmaPlot 13.0 software based on the curve fitting Michaelis–Menten equation: v0 = Vmax[S]/(KM + [S]).

### 4.6. Glycolate Oxidase Activity In Vitro Assay

Glycolate oxidase activity was measured in vitro on protein extracts as described previously [28,53]. Briefly, leaf samples were ground in liquid nitrogen using a TissueLyserII (Qiagen^®^, Retsch, Hilden, Germany) and resuspended in Tris-HCl (50 mM, pH 8). Protein extracts were purified using a NAP-5 Sephadex G-25 DNA grade column (GE Healthcare^®^, Uppsala, Sweden). Enzyme activity was measured in Tris-HCl (50 mM, pH 8) with glycolate (10 mM) and 300 µg of purified total protein by an enzyme-coupled reaction at 30 °C. Glycolate-dependent H_2_O_2_ production was quantified in the presence of 0.4 mM o-dianisidine and 2 units of horseradish peroxidase by measuring the ΔA440 nm using a Varian Cary 50 spectrophotometer.

### 4.7. 35S-Labelled Methionine Incorporation

Seedlings treated with dimethylsulphoxide (DMSO) or 5 µM estradiol for 3 h and transferred to 1 mL of liquid MS supplemented with 1% sucrose containing 50 mCi of 35S-labelled methionine (Perkin-Elmer, Waltham, MA, USA), and incubated for 30 min. Seedlings were rinsed twice for 5 min in MS. Total proteins were extracted in 1 mM sodium ethylenediaminetetraacetate (Na-EDTA), 1 mM MgCl_2_, 25 mM Tris-HCl (pH 7.6), 13.3 mM β-mercaptoethanol and 1 mg/mL antiprotease cocktail (Roche, Meylan, France), subjected to sodium dodecylsulphate polyacrylamide gel electrophoresis (SDS-PAGE) and the methionine incorporation rate was revealed by autoradiography.

### 4.8. RNA Isolation and qRT-PCR

Total RNA was extracted from 100 mg of frozen ground leaves or seedlings using Trizol^®^ reagent (Invitrogen Life Technologies, Saint-Aubin, France). The RNA quality was evaluated by electrophoretic run on 1% agarose gel. First-strand cDNA was synthesized using Superscript reverse transcriptase SSII (Invitrogen, Saint-Aubin, France) from 1 µg of DNase-treated (Invitrogen, Saint-Aubin, France) total RNA in a 20 µL reaction volume. qPCR reactions were performed using SYBR^®^ Selected MasterMix 2x (Applied Biosystem, Villebon Sur Yvette, France), following the manufacturer’s protocol. The cycling conditions consisted of an initial 5 min at 95 C, followed by 40 three-step cycles at 94 °C for 15 s, 60 °C for 30 s, and 72 °C for 30 s. Melting curve analysis was performed after cycle completion to validate amplicon identity. Relative expression levels were calculated following the standard curve-based method [37]. Expression of the Adenosine Phosphoribosyl Transferase 1 (APT1; AT1G27450) reference gene was used for normalization [55]. For each treatment, three biological replicates, corresponding to a pool of 4 leaves from a single plant or to 20 seedlings, were analyzed and each qRT-PCR reaction was carried out in duplicate; the complete experiment was conducted twice independently, and one representative experiment is presented. The gene-specific primers used in this analysis are described in [56].

### 4.9. Detection of ROS

Diaminobenzidine (DAB) staining was used to detect intra-and extracellular hydrogen peroxide (H_2_O_2_). Five-week-old leaves were collected 2 h after inoculation and vacuum-infiltrated with DAB (Sigma Aldrich, Saint Louis, MI, USA) (1 mg/mL, pH 3.7). Leaves were then placed in a wet Petri dish overnight and the staining was stopped at 16 hpi using ethanol to discolor them. 2′,7′-Dichlorofluorescein diacetate (DCFH-DA) staining was used to detect intracellular hydrogen peroxide. A 30 mM DCFH- DA (Sigma Aldrich, Saint Louis, MI, USA) solution was prepared in DMSO and diluted 100 times in deionized water. Inoculated or mock-treated leaves were collected 16 h post-inoculation (hpi) and vacuum-infiltrated with DCFH-DA. Leaves were immediately put on a microscope slide and fluorescence emission was observed under a binocular magnifier with a GFP filter (510 nm).

### 4.10. GOX2::GFP Construct

Plasmid pda09796 containing the *GOX2* gene sequence was obtained from the RIKEN institute. The GFP coding sequence was cloned in 3′ of the *GOX2* coding sequence using the gateway cloning system. The primers used to perform the cloning are detailed in Appendix A (start gox2-2-F; stop gox2-2-R; end gox2-2-R).

### 4.11. Protoplast Production and Transfection

Protoplasts were prepared from 14-day-old *A. thaliana* seedlings stably transformed with a peroxisomal marker [36] as previously described [57]. *A. thaliana* mesophyll protoplasts were transfected with 2.5 µg of the GOX2::GFP construct. The protoplasts were imaged by confocal laser scanning microscopy after 24 h of incubation in the dark at room temperature.

### 4.12. Confocal Laser Scanning Microscopy

Images of fluorescent protoplasts were obtained with a Leica TCS-SP2-AOBS spectral confocal laser scanning microscope equipped with a Leica HC PL APO lbd.BL 20.0x 0.70 water immersion objective. GFP and chloroplasts were excited with the 488 nm line of an argon laser (laser power 40%) while the RFP was excited with the 543 nm line (laser power 30–40%). Fluorescence emission was detected over the range 495 to 540 nm for the GFP construct, 590 to 640 nm for the RFP construct and 670 to 730 nm for chloroplast autofluorescence. Images were recorded and processed using LCS software version 2.5 (Leica Microsystems). Images were cropped using Adobe Photoshop CS2 and assembled using Adobe Illustrator CS2 software (Abode, http://www.abode.com accessed on 1 February 2022).

### 4.13. Metabolite Analysis

Metabolome analysis was performed as previously described [58]. Four pools of 20 seedlings per condition were used for metabolome analysis. Approximately 30 mg of the ground frozen seedling samples was analyzed by an Agilent 7890A gas chromatograph (GC) coupled to an Agilent 5975C mass spectrometer (MS). Standards were injected at the beginning and end of the analysis. Data were analyzed with AMDIS (http://chemdata.nist.gov/mass-spc/amdis/ accessed on 5 December 2013) and QuanLynx software (Waters Corp., Milford, MA, USA).

### 4.14. Structural Model of AtGOX2

The structural model presented in Figure 4b is based on the 3D-structure of *S. oleracea* GOX (PDB 1AL7) [59].

## Figures and Tables

**Figure 1 ijms-23-04224-f001:**
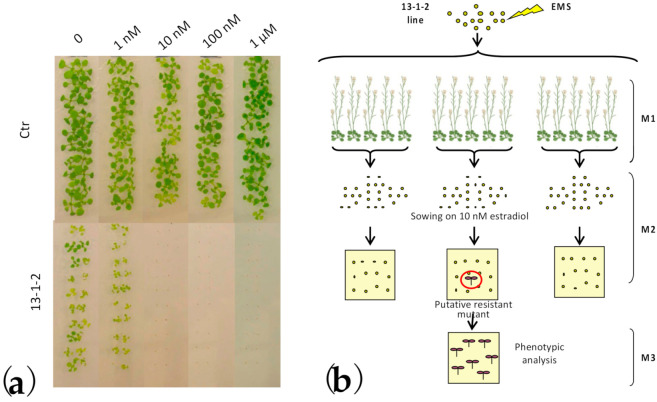
Design of screening for DspA/E-suppressor mutants. (**a**) Seeds of control plants (Ctr) and of a transgenic line with the dspA/E gene under an estradiol-inducible promoter (13-1-2, described in [25]) were sown on MS medium (1% sucrose) with the indicated concentration of estradiol. Pictures were taken two weeks after sowing; (**b**) Outline of DspA/E suppressor mutant selection. The 13-1-2 transgenic line was EMS mutagenized, and the F2 progeny was sown on 10 nM estradiol to identify DspA/Esuppressor mutants. Further molecular characterization was performed to identify mutants in which DspA/E was expressed at least at the same level as in the parental 13-1-2 line. Created with Biorender (https://biorender.com/ accessed on 1 February 2022).

**Figure 2 ijms-23-04224-f002:**
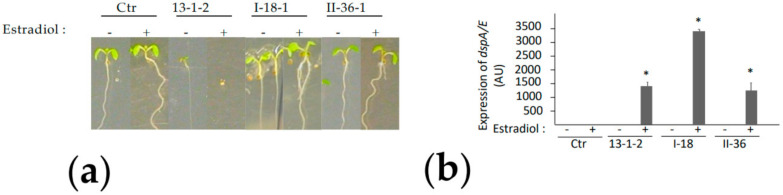
Identification of Arabidopsis mutants suppressing DspA/E-induced toxicity. Seedlings were grown for two weeks on MS medium with 1% sucrose with (+) or without (-) 5 µM estradiol: (**a**) The parental 13-1-2 line does not germinate in the presence of estradiol contrary to the control line. The DspA/E-resistant mutants I-18 and II-36 show a similar phenotype to the control line. The scale bar indicates 0.25 cm; (**b**) Expression of dspA/E gene in mutants is similar or higher to parental line 13-1-2 or higher. Seedlings were treated with 5 µM estradiol (+) or not (-) for 24 h before sampling for RNA extraction and qRT-PCR analysis. Transcript level was normalized to *A. thaliana* APT1 gene. Stars indicate significant difference from control plants (Mann–Whitney, *p*-value < 0.05). Similar results were obtained in at least three independent experiments.

**Figure 3 ijms-23-04224-f003:**
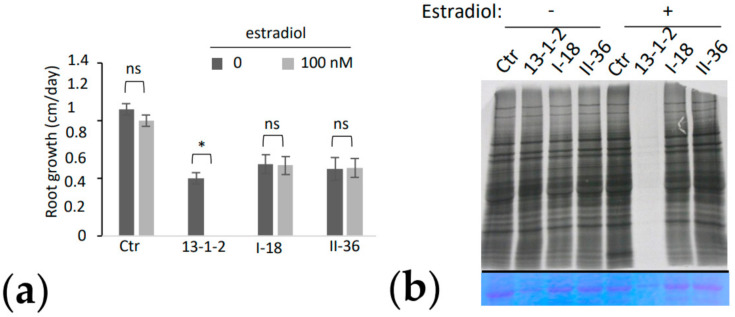
Characterization of phenotypes associated with DspA/E toxicity in suppressor mutants. (**a**) Dose-response of root growth inhibition in the parental 13-1-2 line and in the two selected suppressor mutants. Root growth of mutants is not affected by dspA/E expression in mutants I-18 and II-36. Bars represent standard deviation (SD), n = 20 seedlings; similar results were observed in at least three independent experiments. Stars indicate significant difference to untreated control according to Mann–Whitney (*p*-value < 0.05); (**b**) Translation is not inhibited by DspA/E in suppressor mutants. Ten-day-old seedlings were treated with DMSO (-) or 5 µM estradiol (+) for 3 h and labelled with 35S methionine. Total protein extracts (10 µg per sample) were separated by SDS-PAGE and stained with Coomassie blue (bottom panel); 35S methionine incorporation was detected by autoradiography (top panel). Similar results were obtained for two biological replicates. The picture corresponds to a single gel.

**Figure 4 ijms-23-04224-f004:**
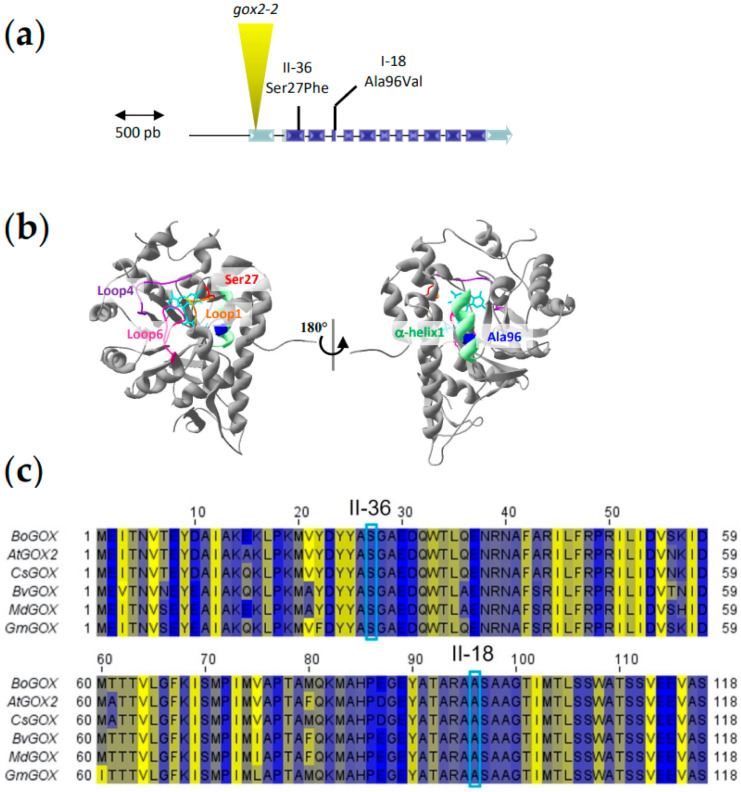
Mutations in the *GOX2* gene responsible for the suppression phenotype (**a**) *GOX2* gene and position of the two-point mutations and of the *gox2-2* T-DNA insertion; (**b**) A structural model of AtGOX2 based on the 3D-structure of Spinacia oleracea GOX (PDB 1AL7); (**c**) Multiple sequence alignments of AtGOX2 with other orthologous protein sequences. Amino acids sequences were retrieved from NCBI, and multiple alignments were performed by “Clustal omega” with default parameters. The alignments were visualized through “Jalview” workbench (version 2.11.1.4) using the beta strand propensity coloring scheme (yellow: high strand propensity, blue: low strand propensity). *At*: *Arabidopsis thaliana*, *Bo*: *Brassica oleracea*, *Bv*: *Beta vulgaris*, *Cs*: *Camelina sativa*, *Md*: *Malus domestica*, *Gs*: *Glycine max*.

**Figure 5 ijms-23-04224-f005:**
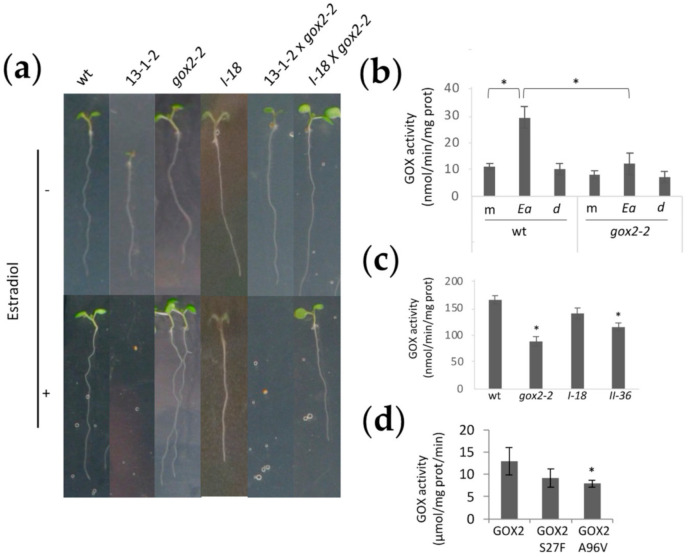
*gox2-2* mutants are allelic to suppressor mutants and mutations in GOX2 suppress DspA/E toxicity in *A. thaliana*. (**a**) Seedlings were grown for 7 days directly on 5 μM estradiol. The control line (wt) and the three parental lines are shown on the left. As expected, the parental 13-1-2 line does not germinate on estradiol while the three other lines do. The F1 progeny of the 13-1-2 x *gox2-2* cross does not germinate on estradiol while the F1 progeny of the I-18 x *gox2-2* cross is able to germinate on 5 μM estradiol. All genotypes were able to germinate on MS without estradiol. Each cross was performed at least in triplicate and 2 to 5 seeds per cross were tested. A representative seedling is shown for the DspA/Eresistant genotypes; (**b**) GOX activity increases following infection of 5-week-old rosette leaves inoculated with wild-type *E. amylovora* (*Ea*). This increase is not observed in response to the *Ea* dspA/E-deficient mutant (**d**) and is not observed in the *A. thaliana gox2-2* mutant. m: mock; *Ea*: wild-type *E. amylovora*; d: *E. amylovora* dspA/E-deficient mutant. For each condition, four pools of three rosette leaves were analyzed; (**c**) GOX activity in the suppressor mutants. GOX activity was measured in vitro. It was strongly reduced in the *gox2-2* mutant and in the II-36 suppressor mutant. In the I-18 we observed a slight reduction that was not significant. For each condition, four pools of 20 seedlings were analyzed; (**d**): GOX activity of GOX2 recombinant proteins bearing mutations found in I-18 (A96V) and II-36 (S27F) mutants. (**b**–**d**): * indicates significant difference with the wild-type (Mann–Whitney, *p*-value < 0.05).

**Figure 6 ijms-23-04224-f006:**
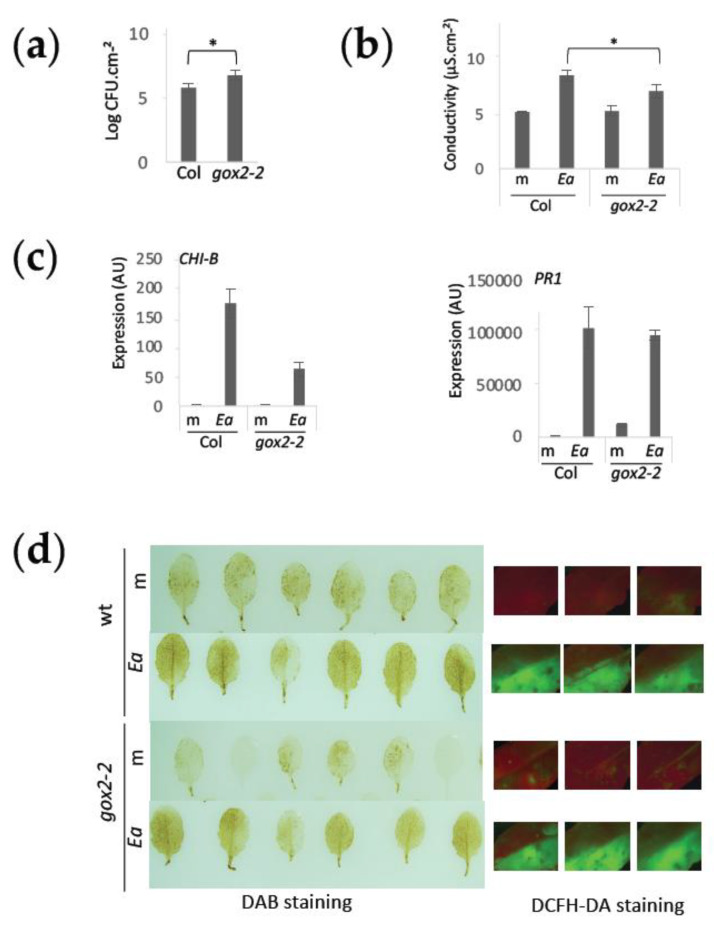
GOX2 is involved in non-host resistance against *E. amylovora*. (**a**) Bacterial titers of wild-type *E. amylovora* in *A. thaliana* leaves 24 hpi. Bacterial titers in the *gox2-2* mutant were 1 log higher than in wild-type leaves; (**b**) Conductivity in *A. thaliana* leaf discs 24 hpi following infection with wild-type *E. amylovora* (*Ea*) or mock-inoculation (m). Conductivity was significantly lower in leaves of the *gox2-2* mutant, indicating lower electrolyte leakage; (**a**,**b**): * indicate significant difference from wild-type plants (Mann–Whitney, *p*-value < 0.05); (**c**) Expression of *CHI-B* and *PR1* genes in mock (m) or *E. amylovora* (*Ea*)-inoculated leaves 24 hpi, relative to the *APT* reference gene, in arbitrary units (AU); (**d**) H_2_O_2_ in leaf tissue detected by DAB (left panel) and DCFH-DA (right panel) staining. An increase in H_2_O_2_ accumulation is observed in leaves 24 h following infection with wild-type *E. amylovora* (*Ea*) in wild-type *A. thaliana* leaves (wt). A similar effect is observed in the *gox2-2* background. For each type of staining a total of 15 leaves per condition were analyzed (representative pictures are shown).

**Figure 7 ijms-23-04224-f007:**
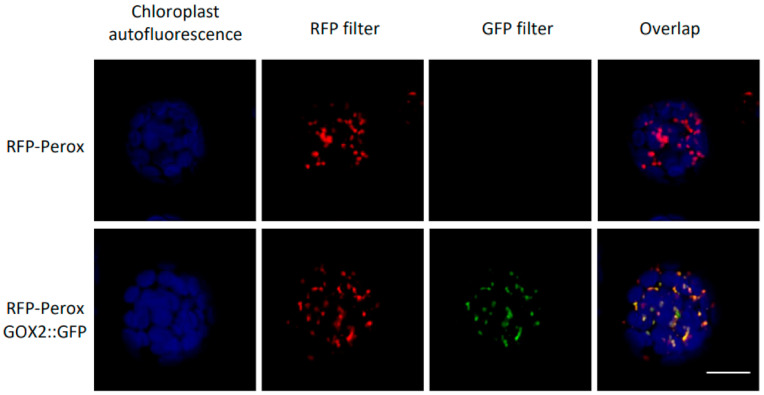
GOX2 is peroxisomal. Transient expression of a GOX2::GFP C-terminal fusion in *A. thaliana* protoplasts from a stable line expressing a RFP-tagged peroxisomal marker [37]. Pictures are single confocal optical sections plane. The RFP, GFP fluorescence and chloroplast autofluorescence are color-coded in red, green and blue, respectively. Scale bar = 30 µm.

**Figure 8 ijms-23-04224-f008:**
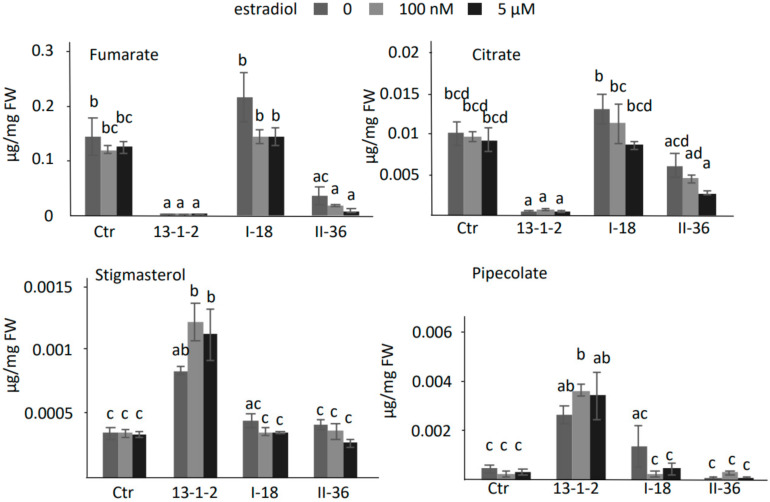
Metabolite accumulation in the 13-1-2 parental line and in the two suppressors. Metabolite accumulation was measured by GC-MS. 10-day old seedlings of the parental 13-1-2 line and of the two suppressor mutants were mock-treated or treated with different concentrations of estradiol (as indicated) to induce DspA/E expression. For each condition, four pools of 20 seedlings were sampled 24 h following treatment. Values correspond to the mean of three biological replicates, bars to the standard error. Differences in mean were tested in a two-way ANOVA test combined with Tukey’s comparison post-hoc test; significantly different means appear with different letters (*p*-value < 0.05).

**Figure 9 ijms-23-04224-f009:**
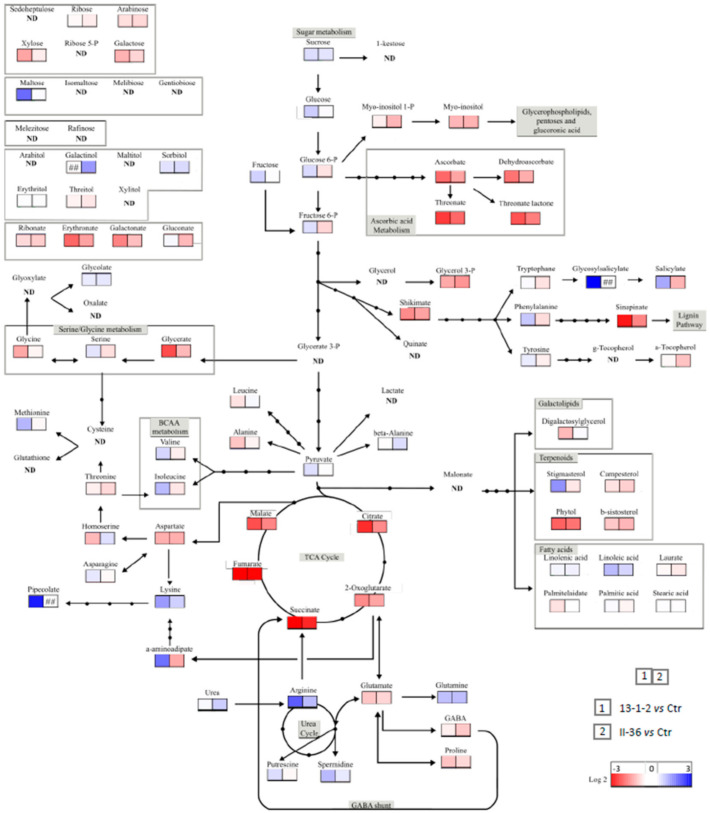
Metabolite accumulation in transgenic lines. DspA/E expression was induced by estradiol treatment in the 13-1-2 line and in the II-36 suppressor and metabolite levels in each of these genotypes was compared to that in the control line without DspA/E. Each square corresponds to log fold accumulation in one condition versus the indicated control condition. Square 1 (**left**): parental 13-1-2 DspA/E line vs. control line; square 2 (**right**): II-36 suppressor vs. control line.

## Data Availability

Not applicable.

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
