# Peer review of "DspA/E-Triggered Non-Host Resistance against E. amylovora Depends on the Arabidopsis GLYCOLATE OXIDASE 2 Gene"

_ijms, 2022, doi:10.3390/ijms23084224_

Round 1

Reviewer 1 Report

DspA/E-triggered non-host resistance against E. amylovora depends on the Arabidopsis GLYCOLATE OXIDASE 2 gene.

Alban Launay, Sylvie Jolivet, Gilles Clément, Marco Zarattini, Younes Dellero, Rozenn Le Hir, Mathieu Jossier, Michael Hodges, Dominique Expert, and Mathilde Fagard.

General comments

The authors described the identification of three suppressor mutants of the type three effector DspA/E. Two of the mutants showed missense mutations in glycolate oxidase 2 (GOX2). A knock-out mutant of GOX2 gene, gox2-2, showed more sensitivity to E. amylovora infection. The results suggest a role of GOX2 or central metabolites in non-host resistance.

This manuscript is clear, but the organization and quality of images should be improved before publication.

Specific comments

  • Abstract, the results on effects of suppressor mutations on metabolite levels should be included in the abstract.
  • Figure 2(a), it is not clear what the low-resolution images were used for. In addition, there are no scale bars on the images.
  • Figure 3(b), were equal amounts of total protein extracts loaded onto each well?
  • Figure 5(b), some of the characters in the sequence alignment image are difficult to see.
  • Figure 6(a), is there any seed in the bottom image of I-18?
  • It is suggested to present the metabolomic results (Figure 4) at the end of the results section as a potential molecular mechanism of the functional role of suppressor mutants and the effects of DspA/E. It is also suggested to include a diagram of TCA cycle that compares the effects of DspA/E and its suppressor on the amounts of each quantified intermediates. The authors may also present the metabolic content results as shown in published references, such as “Zhang, Youjun et al. 2018. The Extra-Pathway Interactome of the TCA Cycle: Expected and Unexpected Metabolic Interactions. Plant Physiology, 177 (3): 966-979”.
  • It is also suggested to include more discussion on potential mechanisms underlying DspA/E and the suppressor mutants. The authors may read the recent review articles, such as “Elina Welchen, at al. 2021. Cross-talk between mitochondrial function, growth, and stress signalling pathways in plants, Journal of Experimental Botany, 72 (11): 4102–4118”.
  • Methods on metabolite analysis using GC-MS are missing in the section of Materials and Methods.

Author Response

Reviewer1 : General comments

The authors described the identification of three suppressor mutants of the type three effector DspA/E. Two of the mutants showed missense mutations in glycolate oxidase 2 (GOX2). A knock-out mutant of GOX2 gene, gox2-2, showed more sensitivity to E. amylovora infection. The results suggest a role of GOX2 or central metabolites in non-host resistance.

This manuscript is clear, but the organization and quality of images should be improved before publication.

    • We thank the reviewer for these positive comments and we took all the comments into account to improve the manuscript.

Specific comments

  • Abstract, the results on effects of suppressor mutations on metabolite levels should be included in the abstract.
    • The result is already in the abstract “These metabolites changes were absent in the suppressor mutants. » We modified the sentence to make it clearer. The sentence is now : « TCA cycle and cell-death associated metabolite levels were respectively increased and reduced in both suppressor mutants compared to the 13-1-2 line. »
  • Figure 2(a), it is not clear what the low-resolution images were used for. In addition, there are no scale bars on the images.
    • Figure 2a shows the phenotype of the mutants as it is mentioned in the manuscript. The low resolution images are due to pixelization of the image when copied into the word document. The low resolution image was replaced with a high resolution image and scale was added to the figure.
  • Figure 3(b), were equal amounts of total protein extracts loaded onto each well?
    • Yes, equal amounts of protein were added to each well (10 micrograms), this information was added to the figure legend
  • Figure 5(b), some of the characters in the sequence alignment image are difficult to see.
    • The Figure was replaced with a new clearer version
  • Figure 6(a), is there any seed in the bottom image of I-18?
    • I-18 is resistant to estradiol-induced DspA/E expression so the seeds germinated to form a seedling. A sentence was added to the legend to make it clearer for the reader.
  • It is suggested to present the metabolomic results (Figure 4) at the end of the results section as a potential molecular mechanism of the functional role of suppressor mutants and the effects of DspA/E. It is also suggested to include a diagram of TCA cycle that compares the effects of DspA/E and its suppressor on the amounts of each quantified intermediates. The authors may also present the metabolic content results as shown in published references, such as “Zhang, Youjun et al. 2018. The Extra-Pathway Interactome of the TCA Cycle: Expected and Unexpected Metabolic Interactions. Plant Physiology, 177 (3): 966-979”.
    • The text was modified as suggested by the reviewer to show the metabolomic data at the end, Figure 4 has become Figure 8 and an extra Figure with the TCA cycle and other major plant metabolism pathways has been added (Figure 9).
  • It is also suggested to include more discussion on potential mechanisms underlying DspA/E and the suppressor mutants. The authors may read the recent review articles, such as “Elina Welchen, at al. 2021. Cross-talk between mitochondrial function, growth, and stress signalling pathways in plants, Journal of Experimental Botany, 72 (11): 4102–4118”.
    • The reference has been added to the discussion as suggested together with some new elements of discussion.
  • Methods on metabolite analysis using GC-MS are missing in the section of Materials and Methods.
    • This information has been added to the “Materials and methods” section

Reviewer 2 Report

This manuscript described identification of GOX2 as a suppressor gene for Erwinia amylovora T3E DspA/E-induced toxicity in Arabidopsis, a nonhost plant, by EMS-mutagenesis on a previously published Arabidopsis transgenic line carrying an inducible DspA/E. Following solid genetic characterization, the authors concluded that the two suppressor mutants harbors causal point mutations that are allelic to a gox2-2 T-DNA KO mutant. While the conclusions are well-supported by the data, the quality of the data presentation could be improved. Below are my detail comments.

In the introduction section, the terminology for type III secretion system (TTSS) and type three effectors (T3Es) need to be consistent. It can be T3SS and T3Es, or TTSS and TTEs, but not mixed.

Several images in Figure 2a are very blurry and therefore are unacceptable. Each photo should have a scale bar.

Figure 3b, it seems all the lanes are not from the same gel. If it is such case, images from different gels should be separated with a gap to prevent misleading the readers to think they are from the same gel.

Figure 4, 6b, 6c and 7a, all the plots are missing axis. The sample sizes for each measurement need to be disclosed in the figure legend.

Figure 5b, the protein structure ID number needs to be disclosed if it is downloaded from PDB database. Alternatively, if the structure is computationally predicted, the detailed method and parameters should be described in the methods section.

Figure 5c, the letters in black are very difficult to read against deep blue background color.

Figure 6a, ideally seedlings from different genotypes grown under the same condition should appear in one photo for comparisons. Otherwise, scale bars are needed.

Figure 6c, why the scale for y-axis is one magnitude higher than in 6b, given that both experiments were measuring GOX activity in Arabidopsis leaf extracts?

Figure 7d, the conclusion that gox2-2 mutant did not affect E. amylovora infection-induced H2O2 is not sufficiently supported by the data, because the data presented is only one leaf photo per sample. A minimally six leaves should be presented for each genotype-treatment combination to substantiate the conclusion. Alternatively, the authors may want to use a quantitative assay.

Related to Figure 8, I am wondering whether subcellular localization of DspA/E and possible colocalization with GOX2 can now be tested using protoplasts isolated from either gox2-2 or the two suppressor mutant plants.

Figure S1, this figure is very confusing. I think some grid with labels are needed to clearly tell the groups of the germinating seedling.

Figure S3, this figure was not referred in the main text at all.

Author Response

Reviewer 2 

This manuscript described identification of GOX2 as a suppressor gene for Erwinia amylovora T3E DspA/E-induced toxicity in Arabidopsis, a nonhost plant, by EMS-mutagenesis on a previously published Arabidopsis transgenic line carrying an inducible DspA/E. Following solid genetic characterization, the authors concluded that the two suppressor mutants harbors causal point mutations that are allelic to a gox2-2 T-DNA KO mutant. While the conclusions are well-supported by the data, the quality of the data presentation could be improved. Below are my detail comments.

 We thank the reviewer for these positive comments and for the suggestions to improve the manuscript.

In the introduction section, the terminology for type III secretion system (TTSS) and type three effectors (T3Es) need to be consistent. It can be T3SS and T3Es, or TTSS and TTEs, but not mixed.

  • The text has been modified to T3E and T3SS

Several images in Figure 2a are very blurry and therefore are unacceptable. Each photo should have a scale bar.

  • The images are replaced by non-blurry ones

Figure 3b, it seems all the lanes are not from the same gel. If it is such case, images from different gels should be separated with a gap to prevent misleading the readers to think they are from the same gel.

  • yes the lanes are from a single gel which was cropped for clarity. The Figure was modified and this information has been added to the figure legend.

Figure 4, 6b, 6c and 7a, all the plots are missing axis. The sample sizes for each measurement need to be disclosed in the figure legend.

  • This information has been added to the legends of Figure 8 (former Figure 4), Figure 5 (former Figure 6) and Figure 6 (former Figure 7). The axis have been added.

Figure 5b, the protein structure ID number needs to be disclosed if it is downloaded from PDB database. Alternatively, if the structure is computationally predicted, the detailed method and parameters should be described in the methods section.

  • The 3D structure is based on the known structure of olareacea GOX. This information has been added to the manuscript.

Figure 5c, the letters in black are very difficult to read against deep blue background color.

  • The Figure has been changed (now Figure 4c)

Figure 6a, ideally seedlings from different genotypes grown under the same condition should appear in one photo for comparisons. Otherwise, scale bars are needed.

  • Scale bars have been added (Now Figure 5a)

Figure 6c, why the scale for y-axis is one magnitude higher than in 6b, given that both experiments were measuring GOX activity in Arabidopsis leaf extracts?

  • Figure 6 is now Figure 5. The differences in GOX levels are likely due to the fact that samples in Figure 5b and 5c do not correspond to the same tissue. In Figure 5b the samples correspond to 5-week old rosette leaves while in Figure 5c the samples correspond to whole seedlings.

Figure 7d, the conclusion that gox2-2 mutant did not affect E. amylovora infection-induced H2O2 is not sufficiently supported by the data, because the data presented is only one leaf photo per sample. A minimally six leaves should be presented for each genotype-treatment combination to substantiate the conclusion. Alternatively, the authors may want to use a quantitative assay.

  • Additional pictures have been added to the figure. Furthermore, we added pictures of a similar experiment performed with another ROS -staining method (DCFH-DA) which have used previously (Launay et al 2016 for example). Bothe methods give similar results for the gox2 mutant.

Related to Figure 8, I am wondering whether subcellular localization of DspA/E and possible colocalization with GOX2 can now be tested using protoplasts isolated from either gox2-2 or the two suppressor mutant plants.

  • In the gox2-2 mutant and the two suppressor mutant plants, GOX2 is likely to be absent so it will not be possible to show any colocalization with DspA/E. Concerning DspA/E localization, the DspA/E protein is unfortunately very labile and very difficult to detect inside plant cells (See Degrave et al 2013 MPP)

Figure S1, this figure is very confusing. I think some grid with labels are needed to clearly tell the groups of the germinating seedling.

  • The Figure was modified to make it clearer.

Figure S3, this figure was not referred in the main text at all.

  • The Figure has been removed

Reviewer 3 Report

This article is still very interesting.

Specific suggestions for improvement as follows:
1. Shorten introduction and discussion

2. Conductivity experiments should increase time points(Generally, it is done within 24 hours, measure once  every hour)

3. Remark the ordinate scale of the figure2(b), figure3(a), and figure7(c), so that those with very small numerical value are seen.

4. The determination of these substances in figure 4 is not described in the materials method.

5. Details such as blue nuclear staining in figure8 are not described in Materials and methods

etc

Author Response

Reviewer 3 : This article is still very interesting.

  • We thank the reviewer for this positive comment.

Specific suggestions for improvement as follows:
1. Shorten introduction and discussion

  • Concerning the discussion, it is rather short and reviewer 1 asked to add some information. On the other hand, I reduced the introduction as suggested.
  1. Conductivity experiments should increase time points(Generally, it is done within 24 hours, measure once  every hour)
  • We agree with the reviewer that it can be interesting to add a kinetics experiment of conductivity, especially if we did not see differences at 24hpi. In our study, there is a clear difference between the two genotypes at the 24h time-point and thus it is sufficient, as shown in other studies in which a single 24h time-point was analyzed.
  1. Remark the ordinate scale of the figure2(b), figure3(a), and figure7(c), so that those with very small numerical value are seen.
  • The Figures have been modified wherever possible
  1. The determination of these substances in figure 4 is not described in the materials method.
  • The method was added to the manuscript
  1. Details such as blue nuclear staining in figure8 are not described in Materials and methods
  • In Figure 8, the organelles appearing in blue correspond to the chloroplasts and not to the nuclei. The autofluorescence of the chloroplasts has been collected as described in the Material and methods. No staining procedure has been used to obtain these images.

Round 2

Reviewer 1 Report

The authors have revised the manuscript as suggested.